# The Mediating Effect of Sleep Quality on the Relationship between Emotional and Behavioral Problems and Suicidal Ideation

**DOI:** 10.3390/ijerph16244963

**Published:** 2019-12-06

**Authors:** Luyao Xiao, Sheng Zhang, Wenyan Li, Ruipeng Wu, Wanxin Wang, Tian Wang, Lan Guo, Ciyong Lu

**Affiliations:** Department of Medical statistics and Epidemiology, School of Public Health, Sun Yat-sen University, Guangzhou 510080, China; xiaoly25@mail2.sysu.edu.cn (L.X.); zhangsh46@mail2.sysu.edu.cn (S.Z.); liwy23@mail2.sysu.edu.cn (W.L.); wurp5@mail2.sysu.edu.cn (R.W.); wgg0808@163.com (W.W.); wangt97@mail2.sysu.edu.cn (T.W.)

**Keywords:** emotional and behavioral problems, suicidal ideation, sleep quality, mediating effect, adolescents

## Abstract

Emotional and behavioral problems in adolescents are associated with suicidal ideation, but different dimensions of problems may be associated with different levels of suicidal ideation. The aim of this large-scale study was to explore the relationship between different dimensions of emotional and behavioral problems and suicidal ideation in Chinese adolescents and to determine whether sleep disorders play a mediating role in the relationship. In total, 20,475 students completed the questionnaire regarding emotional and behavioral problems, sleep quality and suicidal ideation. After adjustment for covariates, total emotional and behavioral difficulties (adjusted odds ratios (AOR) = 1.22, 95% confidence interval (CI) = 1.21–1.23) and sleep disorders (AOR = 4.17, 95% CI = 3.82–4.54) increased the risk of suicidal ideation, while prosocial problems (AOR = 0.91, 95% CI = 0.90–0.93) decreased the risk of suicidal ideation. Sleep quality partially mediated the relationship between emotional and behavioral problems and suicidal ideation. The standardized indirect effects of emotional and behavioral difficulties on suicidal ideation (standardized β estimate = 0.031, 95% CI = 0.020, 0.044) and the effects of prosocial problems on suicidal ideation (standardized β estimate = −0.039, 95% CI = −0.045, −0.035) mediated by sleep quality were statistically significant (*p* < 0.001). Our study indicates that emotional and behavioral problems increase the risk of suicidal ideation. Additionally, sleep quality plays a mediating role in the association between emotional and behavioral problems and suicidal ideation.

## 1. Introduction

Suicide is a significant public health problem and a leading cause of death and disability in China and around the world, especially among adolescents [1]. Suicide seriously endangers the lives and health of young people and causes serious losses to individuals, families and even society. Although the suicide rate in China has decreased in the past decade, it still accounts for the majority of the global suicide rate [2]. Suicidal ideation, a critical part of the suicide process, is an important risk factor for suicide in adolescents [3]. A large cross-sectional study showed that the weighted prevalence of suicidal ideation among Chinese adolescents was 15.7% [4]. Suicide risk is multifactorial, including psychological, social, cultural, behavioral and biological factors. Mental health problems, as an important risk factor for suicide, are predictors of suicidal ideation [5]. Most youth who commit suicide have multiple psychological problems [6]. Adolescence is a critical period of growth, during which adolescents experience various changes. A proportion of adolescents have mental health problems and manifest emotional and behavioral problems [7]. The prevalence of emotional and behavioral problems in Chinese school adolescents is 10~20% [8].

Previous studies have found that emotional and behavioral problems are associated with suicidal ideation. A previous study showed that adolescents who reported suicidal ideation also expressed more emotional and behavioral problems [9]. A nationwide Israeli survey showed that teenagers who attempted suicide had more severe emotional and behavioral problems than those who did not [10]. Bhola et al. found that students who had emotional and hyperactivity difficulties were more likely to report suicidal ideation than those without such difficulties [7]. One study reported that as the level of risk of poor mental health increased, the level of adolescent suicidal ideation also increased [11]. However, these studies considered only a portion of emotional and behavioral problems and did not consider emotional and behavioral problems comprehensively.

The mechanism of the association between emotional and behavioral difficulties and suicidal ideation has not been elucidated. A cohort study found that children with emotional and behavioral problems had more sleep problems [12]. Biological evidence also suggests that sleep is inextricably linked to mood [13]. Moreover, cross-sectional and longitudinal studies showed that poor sleep quality increased the risk of suicidal ideation [14,15,16]. A previous study found that sleep disturbances were associated with psycho-social difficulties, as well as previous suicidal ideation [17]. Therefore, sleep quality may play a mediating role in the association between emotional and behavioral problems and suicidal ideation, while no study has examined this mediating effect.

Thus, we conducted a cross-sectional study in Guangdong Province to evaluate the association between multidimensional emotional and behavioral problems and suicidal ideation in adolescents in school and to explore the mediating effect of sleep quality. We hypothesized that emotional and behavioral problems would be significantly associated with suicidal ideation and that the relationship would be partly mediated by sleep quality.

## 2. Materials and Methods

### 2.1. Study Design and Participants

This was a large-scale cross-sectional study among middle and high school students from Guangdong Province in southern China. The sampling method was three-stage stratified cluster random sampling. In the first stage, according to the per capita gross domestic product (GDP) of Guangdong, the 21 cities/regions in Guangdong Province were divided into three strata: high, medium and low. Two cities were randomly selected as representatives in each stratum. In the second stage, schools in each representative city were divided into three categories: junior high schools, senior high schools, and vocational high schools. In each representative city, we randomly selected 6 middle schools, 4 high schools and 2 vocational high schools according to the proportions of these three types of schools. In the third stage, 2 classes were randomly selected from each grade in the selected schools, and all students in the selected classes were invited to voluntarily participate in our study. A total of 21,019 students were invited to participate in this questionnaire survey, and 20,475 students completed questionnaires and qualified for the survey, for an effective response rate of 97.41%. To ensure the authenticity and effectiveness of the questionnaire and protect students’ privacy, the anonymity of the self-reported questionnaires was guaranteed. The teacher was absent when the students completed the questionnaires in the classroom to avoid potential information bias. The survey was conducted under the supervision of uniformly trained graduate students. The data were collected from 2018 to 2019.

### 2.2. Ethical Statement

This study obtained the ethical approval from the Sun YatSen University School of Public Health Institutional Review Board (ethic code: 2012(20). The approval date was 15 March 2012. After the study procedure had been fully explained, written informed consents were obtained from each participating student who was at least 18 years of age. If the student was under 18 years of age, a written informed consent was obtained from the student’s parents (or legal guardians).

### 2.3. Measures

#### 2.3.1. Emotional and Behavioral Problems

Emotional and behavioral problems were assessed by the self-reported version of the Strengths and Difficulties Questionnaire (SDQ), a multidimensional mental health screening instrument with good reliability and validity that has been translated into many languages, has been implemented in many countries, and is considered a valid, rapid measure of emotional and behavioral problems [18]. The Chinese version of the SDQ has been proven to be valid [19]. Although the behavioral screening tool was originally designed for people aged 11 to 16, a previous study found that it is also applicable to a wider age range (10~19 years) [17]. The SDQ is composed of 25 items, with each item was scored on a 3-point scale (0 = not true, 1 = somewhat true, 2 = certainly true). The 25 items can be divided into 5 subscales: emotional symptoms, conduct problems, hyperactivity-inattention, peer relationship problems and prosocial behavior. The first 4 subscales form a total difficulties score; the higher the score is, the more serious the emotional and behavioral problems. The remaining subscale is the prosocial behavior scale, and the lower the score is, the worse the prosocial behavior problem. In this work, the Cronbach’s alpha for the SDQ was 0.75 and the total difficulties score was 0.72, which proved sufficient internal consistency.

#### 2.3.2. Suicidal Ideation

To assess the suicidal ideation of participants, we asked the following question: “During the past 12 months, how many times did you seriously consider attempting suicide?” The participants chose from two response options: zero or once or more [20].

#### 2.3.3. Sleep Quality

The Pittsburgh Sleep Quality Index (PSQI) is a self-rated questionnaire used to assess participants’ sleep quality and disturbances over the previous month [21]. The PSQI contains 18 items that can be grouped into seven components: (1) subjective sleep quality, (2) sleep latency, (3) sleep duration, (4) habitual sleep efficiency, (5) sleep disturbances, (6) use of sleeping medication, and (7) daytime dysfunction. The score for each component ranges from 0 to 3 points. The PSQI total score ranges from 0 to 21. Higher scores indicate poorer sleep quality. A previous study proved that the PSQI has high validity and reliability and is currently a relatively universal sleep quality screening tool [22]. The PSQI has been proven to be valid and reliable in China [23].

### 2.4. Statistical Analysis

All statistical analyses were performed using SPSS 21.0 (IBM, Chicago, IL, USA) and AMOS 22.0 (IBM Corp., Armonk, NY, USA) software packages. First, descriptive analyses were performed to describe the demographic characteristics, emotional and behavioral problems and sleep quality in adolescents both with and without suicidal ideation; the Rao-Scott 2 and t-tests were utilized to compare the differences between groups. Secondly, multivariable logistic regression models were conducted to examine the relationship between emotional and behavioral problems, sleep quality and suicidal ideation. The odds ratios (ORs), adjusted odds ratios (AORs), and 95% confidence intervals (CIs) were obtained from the logistic regression models. Thirdly, structural equation models using the maximum likelihood (ML) method were utilized to examine the potential mediating role of sleep quality in the association of emotional and behavioral difficulties and prosocial problems with suicidal ideation. The PSQI score and the SDQ score were continuous variables. Suicidal ideation was a categorical variable. The standardized probit coefficients, standardized total effects, and standardized indirect effects were reported, and the bias-corrected 95% CIs were estimated using 2000 bootstrap samples.

## 3. Results

### 3.1. Sample Characteristics

The sample characteristics are shown in Table 1. Among the 20,475 students included in this study, 10,265 (50.1%) were boys, and 10,210 (49.9%) were girls. The mean (SE) age of the students was 15.0 (1.8) years. Overall, 18.2% of the participants (n = 3725) reported having suicidal ideation in the past 12 months. Younger students and female students had higher risks of suicidal ideation. Students with suicidal ideation had higher emotional and behavioral difficulties scores (14.8 vs. 9.6, *p* < 0.001) and global PSQI scores (6.7 vs. 4.5, *p* < 0.001) and lower prosocial behavior problem scores (7.0 vs. 7.3, *p* < 0.001) than those without suicidal ideation.

### 3.2. Associations of Emotional and Behavioral Problems and Sleep Quality with Suicidal Ideation

As shown in Table 2, in the univariable logistic regression models, emotional and behavioral problems and poor sleep quality were associated with past-year suicidal ideation (*p* < 0.001). Furthermore, after adjusting for age and gender, our multivariable logistic regression models revealed that emotional and behavioral problems and poor sleep quality were still associated with past-year suicidal ideation. Total difficulties (AOR = 1.22, 95% CI = 1.21–1.23) and sleep disorders (AOR = 4.17, 95% CI = 3.82–4.54) increased the risk of suicidal ideation, while prosocial problems (AOR = 0.91, 95% CI = 0.90–0.93) decreased the risk of suicidal ideation.

### 3.3. Mediating Effects of Sleep Quality

The structural equation models using the ML method were utilized to examine the potential mediating role of sleep quality in the association of emotional and behavioral difficulties with suicidal ideation (Figure 1). Structural equation models utilizing the ML method were used to examine the potential mediating role of sleep quality in the association of prosocial problems with suicidal ideation (Figure 2). The standardized path coefficients, standardized indirect effects and standardized total effects for past-year suicidal ideation are shown in Table 3 and Table 4. The direct standardized path coefficients of the effects of emotional and behavioral difficulties on sleep quality (standardized β estimate = 0.602, 95% CI = 0.589, 0.615), the effects of prosocial problems on sleep quality (standardized β estimate = −0.124, 95% CI = −0.139, −0.110) and the effects of sleep quality on suicidal ideation (standardized β estimate = 0.321, 95% CI = 0.308, 0.335) were statistically significant (*p* < 0.001). The standardized indirect effects of emotional and behavioral difficulties on suicidal ideation (standardized β estimate = 0.031, 95% CI = 0.020, 0.044) and the effects of prosocial problems on suicidal ideation (standardized β estimate = −0.039, 95% CI = −0.045, −0.035) mediated by sleep quality were statistically significant (*p* < 0.001).

## 4. Discussion

Although suicidal ideation does not necessarily lead to suicidal behavior, it is the most sensitive predictor of suicide [24]. Early detection of suicidal ideation and early intervention among adolescents can be achieved and would be very helpful in preventing suicidal behavior. Our study found that suicidal ideation is common in Chinese adolescents. The prevalence of suicidal ideation in this study was 18.2%, which is higher than that observed in our previous national survey (15.7%) conducted in 2014–2015 [4]. The causes of the increased prevalence rates may be related to the different sample sizes and increasing academic stress in China [25]. Consistent with previous studies, older students had lower risks of suicidal ideation and female students had higher risks of suicidal ideation [26,27].

A full understanding of the relationship between mental health problems and suicidal ideation has important implications for planning suicide ideation prevention measures. The results of the multivariate regression analysis showed that all SDQ subscales were correlated with suicidal ideation in our study. Emotional and behavioral difficulties (including emotional symptoms, conduct problems, hyperactivity, and peer problems) increase the risk of suicidal ideation, while prosocial problems decrease the risk of suicidal ideation. The OR of reporting suicidal ideation varied according to adolescents’ different dimensions of psychological problems. The present findings suggest that emotional symptoms, conduct problems, and hyperactivity were significantly associated with suicidal ideation. Peer problems and prosocial problems had less impact on suicidal ideation. After adjusting for gender and age, the results remained stable. Our results are similar to those of a nationwide Israeli survey, which indicated that a self-reported version of SDQ was useful in the detection of suicidal adolescents. Their results showed that all self-reported SDQ problem subscales were associated with suicidal ideation, while the prosocial self-scale seemed insensitive to the identification of suicidal adolescents [10]. The SDQ emotional symptoms subscale includes items for feelings of sickness, worry, depression, nervousness, and fear. These factors, especially depression, are significant risk factors for suicidal ideation [28,29,30]. Consistently with a previous study, our study suggests that adolescents with suicidal ideation have more conduct problems than those without suicidal ideation [31]. Early identification and prevention of symptoms of hyperactivity and inattention among youth is also important. A cohort study reported that hyperactivity-inattention predicted suicidal behaviors [32]. In our study, peer relationship problems among adolescents were associated with their self-reported suicidal ideation; thus, good interpersonal relationships may play an important role in promoting students’ senses of belonging in school [33]. There has been little research on the relationship between prosocial problems and suicidal ideation. In our study, we found a weak link between the two variables. These findings suggest that mental health problems are important predictors of suicidal ideation in adolescents and that early identification and appropriate interventions for psychiatric disorders are needed to reduce the prevalence of suicidal ideation. In addition, we need to take different approaches to different types of mental health problems.

Our results reveal that emotional and behavioral difficulties were positively correlated with poor sleep quality, while prosocial problems were negatively correlated with poor sleep quality. Similarly, a study showed that sleep disturbances were associated with emotional/behavioral problems in China and Japan [34]. Consistently with previous studies, teenagers with sleep disorders reported more suicidal ideation [35,36].

The structural equation model shows that sleep quality partially mediated the relationship between emotional and behavioral problems and suicidal ideation and that sleep quality weakened the association between the two variables. Chinese middle school students sleep less because of the pressure of high school and college entrance exams. One study showed that 14.6% of Chinese students had to wake up earlier than 6:00 and that the vast majority (94.4%) slept less than 8 h per night [37], which is not good for their physical and mental health. Sleep disorders are common in adolescents, especially in those with emotional and behavioral problems; Van Dyk found that sleep and mental health symptoms were highly related at the daily level [38]. Some emotional and behavioral problems, such as depression, anxiety or hyperactivity, can lead to sleep disturbances [39,40,41]. Clinical evidence also suggests that sleep and emotion interact; nearly all psychiatric and neurological disorders expressing sleep disruption display corresponding symptoms of affective imbalance [13]. Adolescents with emotional and behavioral problems may have sleep disorders, which are a proven risk factor for suicidal ideation [16,42,43]. Sleep disorders play a mediating role in the relationship between emotional and behavioral problems and suicidal ideation. Previous studies have linked emotional and behavioral problems to suicidal ideation, but the mechanism of the relationship has not been clarified. Our findings provide a new hypothesis for understanding the process of suicidal ideation in adolescents with emotional and behavioral problems, which may help in the early detection and prevention of suicidal ideation. Thus, sleep quality can be a marker of suicidal risk. This study suggests that improving sleep quality, especially in adolescents with emotional and behavioral problems, is important in preventing suicidal ideation.

Our results have implications for the prevention of suicidal ideation. Early identification and treatment of adolescents’ emotional and behavioral problems and improvement of their sleep quality are important for preventing suicidal ideation. Considering these findings, the following recommendations can be taken to reduce the risk of suicidal ideation. First, government and public health departments should establish a nationwide suicide monitoring system (such as the Youth Risk Behavior Surveillance System) to monitor suicidal ideation in adolescents. Secondly, schools should regularly screen students for mental health status and conduct mental health education courses. Thirdly, parents and teachers should pay close attention to teenagers’ emotional and behavioral problems and communicate with them in a timely manner. Fourthly, psychologists, physicians and related clinical workers should provide preventive interventions and treatment plans that correspond to different types of emotional and behavioral problems. Fifth, effective measures to improve sleep health for those adolescents with emotional behavior problems should be taken.

To our knowledge, this is the first large-scale study to explore the relationship between emotional and behavioral problems, sleep quality and suicidal ideation and to examine the mediating effect of sleep quality among Chinese adolescents. In addition, we comprehensively explored multiple dimensions of psychological problems and used all five subscales of the SDQ among adolescents. However, there are several notable limitations in our study. First, because our study was a cross-sectional study, we could not infer any causal relationships between emotional and behavioral problems, sleep disorders, and suicidal ideation. Secondly, our sample included only middle school students and did not include teenagers who were absent from school on the day of the survey or who had dropped out of school, who may be more likely to have emotional and behavioral problems, sleep problems and suicidal ideation. Third, the self-reported questionnaire may have recall bias, which may affect the authenticity of the results. Fourthly, suicidal ideation was measured by single items instead of standard scales in this study. Future studies should use standard scales to measure suicidal ideation. Fifth, although our multivariate logistic regression and structural equation models controlled for age and gender, there may be other possible confounding factors that affect emotional and behavioral problems and suicidal ideation, such as quality of life and bullying [44,45]. Sixth, although this was a large-scale study including 20,475 students, the participants were recruited only from Guangdong Province, China, which may limit the extrapolation of the results. Seventh, although our logistic regression model and SEM model were adjusted covariates of age and gender, other potential confounding factors (e.g., drug abuse and post-traumatic stress disorders) that might affect sleep quality and suicidal ideation were not included in our study. These factors need to be considered in our future studies.

## 5. Conclusions

Our study indicates that emotional and behavioral problems increase the risk of suicidal ideation among adolescents; the scores for emotional and behavioral problems were positively correlated with suicidal ideation, whereas the scores for prosocial problems were negatively correlated with suicidal ideation. We found that sleep quality partly mediated the association between emotional and behavioral problems and suicidal ideation. For students with suicidal ideation, especially those with emotional and behavioral problems, we should take timely intervention measures to prevent the occurrence of suicidal behaviors. In addition, we should pay close attention to adolescents’ sleep quality and take appropriate treatment measures for adolescents with sleep disorders to improve their sleep health.

## Figures and Tables

**Figure 1 ijerph-16-04963-f001:**
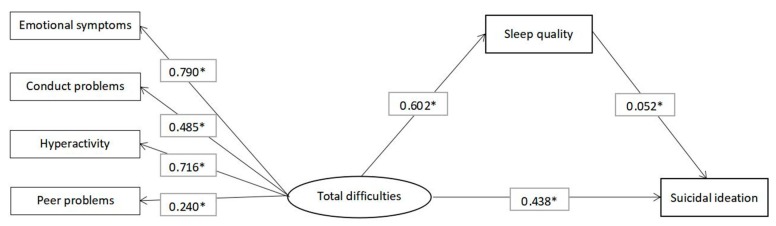
Structural equation model of the relationship between total difficulties, sleep quality and suicidal ideation. Note: Gender and age were the covariates for each variable. * *p* < 0.001.

**Figure 2 ijerph-16-04963-f002:**
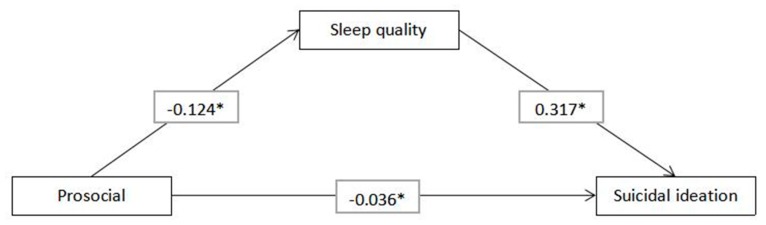
Structural equation model of the relationship between prosocial problems, sleep quality and suicidal ideation. *Note*: Gender and age were the covariates for each variable. * *p* < 0.001.

**Table 1 ijerph-16-04963-t001:** Characteristics of participants with and without suicidal ideation (n = 20,475).

Variables	Total (%)	Suicidal Ideation	No Suicidal Ideation	*χ^2^*/*t*	*p*
Total, N (%)	20,475 (100.0)	3725 (18.2)	16,750 (81.8)		
Gender				363.871	<0.001
Male	10,265 (50.1)	1341 (36.0)	8924 (53.3)		
Female	10,210 (49.9)	2384 (64.0)	7826 (46.7)		
Age, mean (SD)	14.95 (1.78)	14.86 (1.71)	14.98 (1.79)	−3.891	<0.001
SDQ, mean (SD)					
Total difficulties	10.58 (5.18)	14.83 (5.43)	9.63 (4.62)	60.133	<0.001
Emotional symptoms	2.29 (2.25)	4.20 (2.55)	1.87 (1.94)	62.455	<0.001
Conduct problems	1.90 (1.37)	2.57 (1.58)	1.75 (1.27)	33.914	<0.001
Hyperactivity/inattention	3.39 (2.14)	4.78 (2.17)	3.08 (2.01)	46.017	<0.001
Peer relationship problems	2.99 (1.50)	3.28 (1.63)	2.93 (1.46)	13.409	<0.001
Prosocial behavior	7.25 (2.19)	6.98 (2.14)	7.31 (2.19)	−8.514	<0.001
PSQI, mean (SD)					
Subjective sleep quality	1.13 (0.72)	1.44 (0.75)	1.06 (0.70)	29.512	<0.001
Sleep latency	0.65 (0.80)	0.95 (0.92)	0.58 (0.76)	25.764	<0.001
Sleep duration	0.51 (0.72)	0.74 (0.83)	0.46 (0.69)	21.819	<0.001
Habitual sleep efficiency	0.21 (0.64)	0.29 (0.66)	0.19 (0.55)	9.361	<0.001
Sleep disturbances	0.79 (0.57)	1.07 (0.60)	0.72 (0.54)	34.473	<0.001
Use of sleeping medication	0.02 (0.20)	0.07 (0.38)	0.01 (0.13)	15.781	<0.001
Daytime dysfunction	1.55 (0.92)	2.09 (0.81)	1.43 (0.90)	41.517	<0.001
PSQI total score	4.86 (2.72)	6.65 (2.87)	4.46 (2.52)	46.695	<0.001

SDQ: Strengths and Difficulties Questionnaire; PSQI: Pittsburgh Sleep Quality Index; SD: standard deviation.

**Table 2 ijerph-16-04963-t002:** Association between emotional and behavioral problems, sleep quality and suicidal ideation.

Variable	Suicidal Ideation			
	OR (95% CI)	*p* Value	AOR (95% CI)	*p* Value
Total difficulties *	1.22 (1.21, 1.23)	<0.001	1.22 (1.21, 1.23)	<0.001
Emotional symptoms *	1.54 (1.51, 1.56)	<0.001	1.52 (1.50, 1.55)	<0.001
Conduct problems *	1.49 (1.46, 1.53)	<0.001	1.52 (1.49, 1.56)	<0.001
Hyperactivity/inattention *	1.46 (1.44, 1.49)	<0.001	1.47 (1.44, 1.49)	<0.001
Peer relationship problems *	1.17 (1.14, 1.19)	<0.001	1.19 (1.16, 1.22)	<0.001
Prosocial problems *	0.93 (0.92, 0.95)	<0.001	0.91 (0.90, 0.93)	<0.001
Sleep disorders				
No	1 (reference)		1 (reference)	
Yes	4.00 (3.68, 4.35)	<0.001	4.17 (3.82, 4.54)	<0.001

* 1-point score increase. OR: odds ratio; AOR: adjusted odds ratio; CI: confidence interval.

**Table 3 ijerph-16-04963-t003:** Mediating effect of sleep quality on the association between emotional and behavioral difficulties and suicidal ideation.

Variables	β (95% CI)	*p*
Path		
Total difficulties → Sleep quality	0.602 (0.589, 0.615)	<0.001
Sleep quality → Suicidal ideation	0.052 (0.033, 0.073)	<0.001
Total difficulties → Suicidal ideation	0.438 (0.416, 0.459)	<0.001
Standardized effect		
Indirect effect	0.031 (0.020, 0.044)	<0.001
Total effect	0.469 (0.454, 0.483)	<0.001

CI: confidence interval.

**Table 4 ijerph-16-04963-t004:** Mediating effect of sleep quality on the association between prosocial problems and suicidal ideation.

Variables	β (95% CI)	*p*
Path		
Prosocial problems → Sleep quality	−0.124 (−0.139, −0.110)	<0.001
Sleep quality → Suicidal ideation	0.317 (0.303, 0.331)	<0.001
Prosocial problems → Suicidal ideation	−0.036 (−0.050, −0.024)	<0.001
Standardized effect		
Indirect effect	−0.039 (−0.045, −0.035)	<0.001
Total effect	−0.075 (−0.089, −0.062)	<0.001

CI: confidence interval.

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
