# Peer review of "The Mediating Effect of Sleep Quality on the Relationship between Emotional and Behavioral Problems and Suicidal Ideation"

_ijerph, 2019, doi:10.3390/ijerph16244963_

Round 1
Reviewer 1 Report
. Authors investigated the role of sleep disorders in mediating the effect of emotional and behavioral problems on suicidal ideation in Chinese adolescent population. The manuscript is well written. I only have one major concern: Sleep problems can happen from multiple reasons and may not be the primary cause. One of the most common factors is drug abuse especially in adolescents. In fact, drug of abuse such as opioid and alcohol, itself is an independent determinant factor for suicidal ideation. Another important factor is post-traumatic stress disorders (PTSD). Did authors set any inclusion or exclusion criteria to determine if suicidal ideation is only because of sleep problems or due to other factors causing both sleep problems and suicidal ideation.
Author Response
Point 1: Authors investigated the role of sleep disorders in mediating the effect of emotional and behavioral problems on suicidal ideation in Chinese adolescent population. The manuscript is well written. I only have one major concern: Sleep problems can happen from multiple reasons and may not be the primary cause. One of the most common factors is drug abuse especially in adolescents. In fact, drug of abuse such as opioid and alcohol, itself is an independent determinant factor for suicidal ideation. Another important factor is post-traumatic stress disorders (PTSD). Did authors set any inclusion or exclusion criteria to determine if suicidal ideation is only because of sleep problems or due to other factors causing both sleep problems and suicidal ideation.
Response 1: Thank you for your kind suggestion. Indeed, there may be other confounding factors for sleep quality and suicidal ideation. Drug abuse and post-traumatic stress disorder as you mentioned are important confounding factors. In fact, there were questions about drug abuse, such as opioid use, in our questionnaires, but because high school students are under strict supervision in China, the number of drug abusers is very small. The relevant population was excluded when included in the analysis. PTSD was not covered in our questionnaires, so we didn't take into account the impact, which is a weakness of our study. Your suggestions are very important, we will take these effects into account in our future studies. We have added this deficiency to the limitations of our study in this paper (please see page 8, lines 283-286).

Reviewer 2 Report
This study aims to explore the relationship between emotional and behavioral problem and suicidal ideation and the mediating effect of sleep quality on the relationship. The hypothesis, design, results and argument were properly presented in a well-written structure. The authors have used a large sample and appropriate measures and statistical analysis. Although the findings are not much of innovation, this is a well-organized research. I only have some minor comments as followed.
Line 53, p. 2. I suggest putting Bhola “et al.” instead of Bhola only. The authors may want to provide some reference of Chinese versions of SDQ and PSQI and the reliability and validity of the Chinese versions. Section 2.4, p.3. The authors treated emotional and behavioral difficulties as categorical variables. But according to the introduction in section 2.3.1 (3-point scales) and table 1 (presented by mean and SD), the difficulties scores seem to be continuous variables. Table 1. Suggest adding “,N (%)” after “Total” and “Gender” in the first column and eliminate “N (%)” in the first row. Section 3.3 and table 3 and 4. The value of β (0.321) of the effects of sleep quality on suicidal ideation are different from the values presented in figure 1 and 2 ( 0.052 and 0.317 respectively).
Author Response
Point 2: This study aims to explore the relationship between emotional and behavioral problem and suicidal ideation and the mediating effect of sleep quality on the relationship. The hypothesis, design, results and argument were properly presented in a well-written structure. The authors have used a large sample and appropriate measures and statistical analysis. Although the findings are not much of innovation, this is a well-organized research. I only have some minor comments as followed.
Line 53, p. 2. I suggest putting Bhola “et al.” instead of Bhola only. The authors may want to provide some reference of Chinese versions of SDQ and PSQI and the reliability and validity of the Chinese versions. Section 2.4, p.3. The authors treated emotional and behavioral difficulties as categorical variables. But according to the introduction in section 2.3.1 (3-point scales) and table 1 (presented by mean and SD), the difficulties scores seem to be continuous variables. Table 1. Suggest adding “,N (%)” after “Total” and “Gender” in the first column and eliminate “N (%)” in the first row. Section 3.3 and table 3 and 4. The value of β (0.321) of the effects of sleep quality on suicidal ideation are different from the values presented in figure 1 and 2 ( 0.052 and 0.317 respectively).
Response 2: Thank you for carefully and patiently reviewing our manuscript. We apologize for some expression errors in our original manuscript. First, we have put Bhola “et al.” instead of Bhola only (please see page 2, lines 54). Second, as suggested, we have provided references to prove that Chinese versions of SDQ and PSQI are valid and reliable (please see page 3, lines 103 and page 3, lines 125). Third, the PSQI score and the SDQ score were continuous variables. Suicidal ideation were was categorical variables. We have corrected it in the manuscript (please see page 3-4, lines 137-138). Fourth, according to your suggestion, Table 1. “,N (%)” has been added after “Total” and “Gender” in the first column and “N (%)” has been eliminated in the first row (please see page 4, lines 150, Table 1). Fifth, we have showed the correct β values and 95% CI that are affected by the mediating effect in Table 3 and Table 4 (please see page 6, lines 186-188, Table 3 and Table 4).
